# Structural dynamics of the human Orai1 channel revealed by cryo-electron microscopy

Yiming Zhang[1,2☯], Yuan Wang[3,4,5☯], Jindou Liu[1☯], Weiwei Bei[2☯], Hongkun Wang[1], Junli Wang[6]*, Lei Chen[3,4,5,7]*, Youjun Wang[1,8]*

1 Beijing Key Laboratory of Gene Resource and Molecular Development, College of Life Sciences, Beijing Normal University, Beijing, China, 2 National Laboratory of Biomacromolecules, CAS Center for Excellence in Biomacromolecules, Institute of Biophysics, Chinese Academy of Sciences, Beijing, China, 3 State Key Laboratory of Membrane Biology, College of Future Technology, Institute of Molecular Medicine, Beijing Key Laboratory of Cardiometabolic Molecular Medicine, Peking University, Beijing, China, 4 Peking-Tsinghua Center for Life Sciences, Peking University, Beijing, China, 5 Academy for Advanced Interdisciplinary Studies, Peking University, Beijing, China, 6 College of Life and Environmental Sciences, Minzu University of China, Beijing, China, 7 National Biomedical Imaging Center, Peking University, Beijing, China, 8 Key Laboratory of Cell Proliferation and Regulation Biology, Ministry of Education, College of Life Sciences, Beijing Normal University, Beijing, China

☯ These authors contributed equally to this work.
* junliwang@muc.edu.cn (JW); chenlei2016@pku.edu.cn (LC); wyoujun@bnu.edu.cn (YW)

## Abstract

The pore-forming Orai1 protein is an essential component of store-operated calcium entry (SOCE), a process vital to diverse cellular and physiological functions. Mutations in human Orai1 cause severe immunodeficiencies and myopathies, yet structural insights have remained largely elusive. To address this, we studied the structure of detergent-solubilized human Orai1 (hOrai1) by cryo-electron microscopy. While the overall resolution is moderate, the reconstructed map confirms a conserved hexameric architecture and enables assignment of transmembrane helices. We observed profound structural heterogeneity, with particles adopting both C6- and C2-symmetric conformations, indicative of dynamic rearrangements. This study establishes a framework for future structural and mechanistic studies of hOrai1.

## Main

Store-operated Ca²⁺ entry (SOCE) constitutes a fundamental and evolutionarily conserved $Ca^{2+}$ signaling pathway in metazoans [1,2]. Mediated by endoplasmic reticulum (ER) $Ca^{2+}$ sensor STIM1 and plasma membrane (PM) channel pore Orai1, authentic SOCE is a critical signaling event in numerous physiological processes, including immune cell activation [3] and gene transcription [4,5]. Mutations in the hOrai1 gene, leading to either loss-of-function or gain-of-function, are directly linked to severe human diseases such as immunodeficiencies and myopathies, underscoring its non-redundant physiological importance [4]. Therefore, elucidating the precise

**Data availability statement:** All relevant data are within the manuscript and its Supporting Information files.

**Funding:** This work was supported by the National Natural Science Foundation of China (grant Nos. 92254301 and W2411015 to Y.W., 32225027 to L.C.), and the Center for Life Sciences (CLS, to L.C.). The funders had no role in study design, data collection and analysis, decision to publish, or preparation of the manuscript. National Natural Science Foundation of China: https://www.nsfc.gov. cn/. Center for Life Sciences: http://www.cls. edu.cn/.

**Competing interests:** The authors have declared that no competing interests exist.

molecular architecture of hOrai1 is essential for understanding its unique biophysical properties and for developing targeted therapeutic strategies.

Significant progress has been made in understanding Orai channel structure, primarily through studies of its *Drosophila melanogaster* ortholog. Landmark crystallographic and cryo-EM studies have revealed that *Drosophila* Orai (dOrai) assembles as a hexamer, with each subunit contributing four transmembrane helices to form a central pore [6–9]. These structures have provided invaluable insights into the channel's closed (quiescent) state and, more recently, into constitutively active mutant conformations that mimic the open state, shedding light on potential gating and ion permeation mechanisms [7–9]. However, despite considerable sequence conservation in the transmembrane regions, direct structural information on hOrai1, the isoform critically associated with human disease, remains scarce [10]. The lack of a high-resolution structure for hOrai1 limits our ability to interpret disease-associated mutations in their native context and to assess isoform-specific functional differences.

To bridge this gap, here we report the structural analysis of hOrai1. Although the achieved resolution is moderate, our structure confirms the conserved hexameric architecture in the human channel and allows for the unambiguous assignment of key pore-lining helices. This model serves as a foundational reference for future structural and mechanistic studies targeting this vital calcium channel.

## Functional analysis and purification of hOrai1

To alleviate the protein degradation, we generated an N-terminally truncated hOrai1 construct (residues 60–301) C-terminally fused with a Strep-tag (hOrai1-ΔN-Strep). To assess whether this variant is able to mediate SOCE at the level comparable to that of full length hOrai1, we performed $Ca^{2+}$ imaging in HEK 293 cells stably expressing STIM1-YFP (HEK STIM1 cells) [11], together with the genetically encoded $Ca^{2+}$ indicator R-GECO1.2 [12,13]. SOCE was evaluated using a standard $Ca^{2+}$-add-back assay [14]: ER $Ca^{2+}$ stores were depleted by incubating cells for 10-min in nominally $Ca^{2+}$ free solution containing 1 μM thapsigargin (an ER $Ca^{2+}$-ATPase inhibitor), followed by the addition of 1 mM extracellular $Ca^{2+}$ to trigger $Ca^{2+}$ influx via SOCE. The results showed that HEK STIM1 cells transiently co-expressing hOrai1-ΔN-Strep and R-GECO1.2 exhibited SOCE signals similar to those expressing wild type hOrai1 (Fig 1A), indicating that the N-terminal deletion does not impair SOCE function. This observation is consistent with previous reports [2,15,16].

We next expressed hOrai1-ΔN-Strep and solubilized it in LMNG-CHS mixed micelles. The protein was purified via Streptactin affinity chromatography and size-exclusion chromatography (SEC). The protein showed a monodisperse peak on SEC (Fig 1B). Analysis of the peak fractions by SDS-PAGE, followed by Coomassie blue staining and western blot, revealed a major band of hOrai1 (Fig 1C), indicating high protein purity suitable for further structural studies. Native PAGE analysis further indicated that the purified hOrai1-ΔN-Strep protein assembles as an oligomeric complex (Fig 1D). Uncropped and unadjusted blot/gel images corresponding to Fig 1C and Fig 1D are provided in S1 File. SEC-purified hOrai1 was used for cryo-EM sample

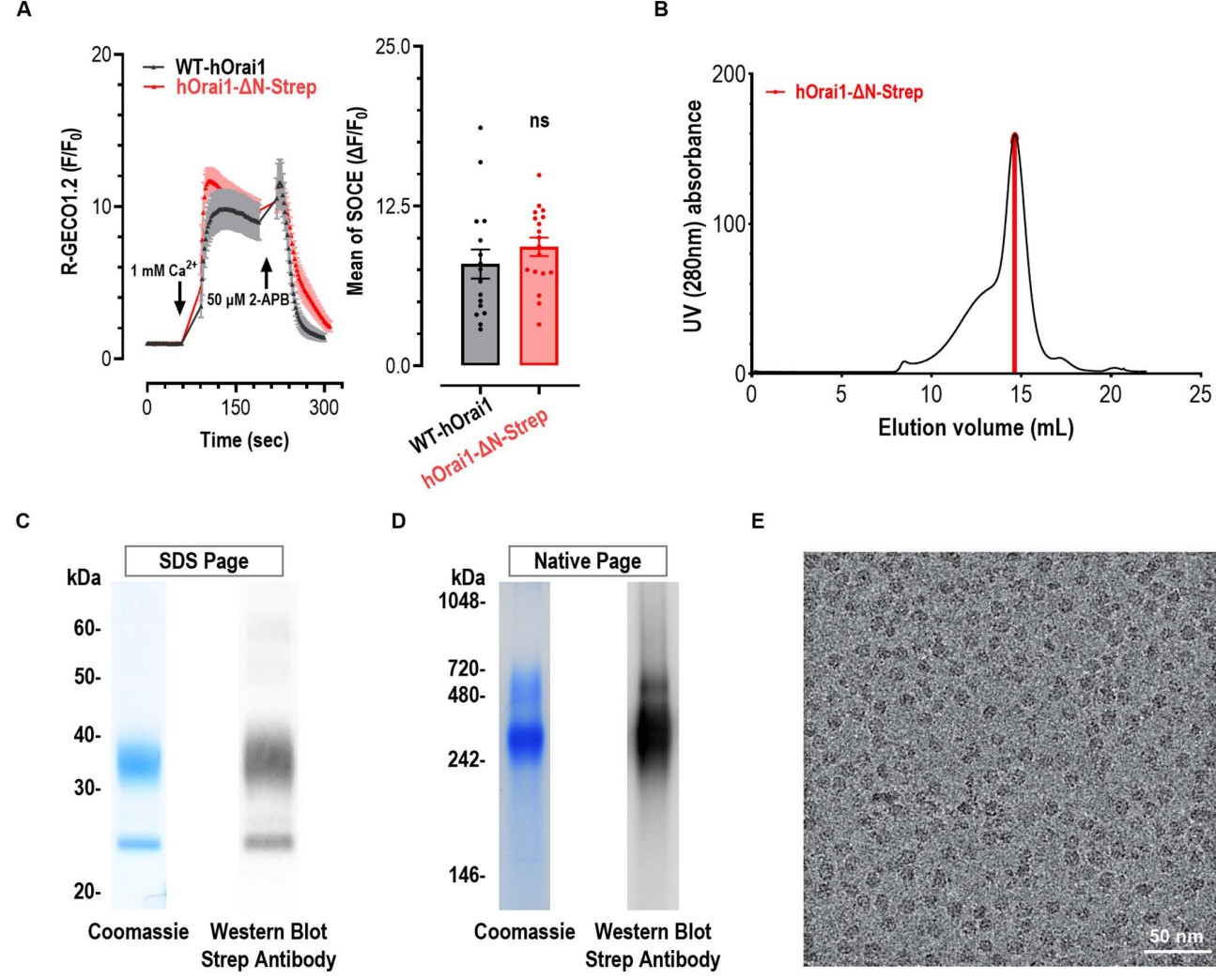

**Fig 1. Function and purification of hOrai1$_{60-301}$-Strep.** (A) Thapsigargin (1 μM)-induced SOCE responses indicated by R-GECO1.2 in HEK STIM1-YFP stable cells. R-GECO1.2 was transiently co-expressed with either wide type hOrai1 or hOrai1$_{60-301}$-Strep (hOrai1-ΔN-Strep). Before recordings, cells were incubated in nominally Ca$^{2+}$ free solution containing 1 μM thapsigargin (TG) to allow full depletion of ER Ca$^{2+}$ stores. TG was present throughout the recordings. Left, typical traces; right, statistics. (n = 3, Student t-test, p = 0.3157). (B) SEC elution profiles of the hOrai1-ΔN-Strep. Protein from the red area was used for Cryo-EM sample preparation. (C) The pooled fractions were analyzed by SDS-PAGE with Coomassie blue staining and Western blot. (D) The pooled fractions were analyzed by Native-PAGE, with detection by Coomassie blue staining and Western blot. (E) A representative cryo-EM micrograph of purified hOrai1.

preparation. The cryo-EM micrographs showed well-dispersed protein particles embedded in vitreous ice (Fig 1E), and a large-scale cryo-EM data was collected using a 300 kV Titan Krios microscope.

## hOrai1 shows dynamic structures

Two-dimensional classification and averaging yielded projections of hOrai1 in various orientations. To assess the structural heterogeneity of the sample from 2D average (Fig 2), we generated a simulated projection map of dOrai in the unlatched closed conformation with C6 symmetry from previously determined X-ray structure (PDB ID: 6BBG) [7] (Fig 2A-B). In the vertical orientation, the projection of dOrai exhibited distinctive features resembling a regular hexagon

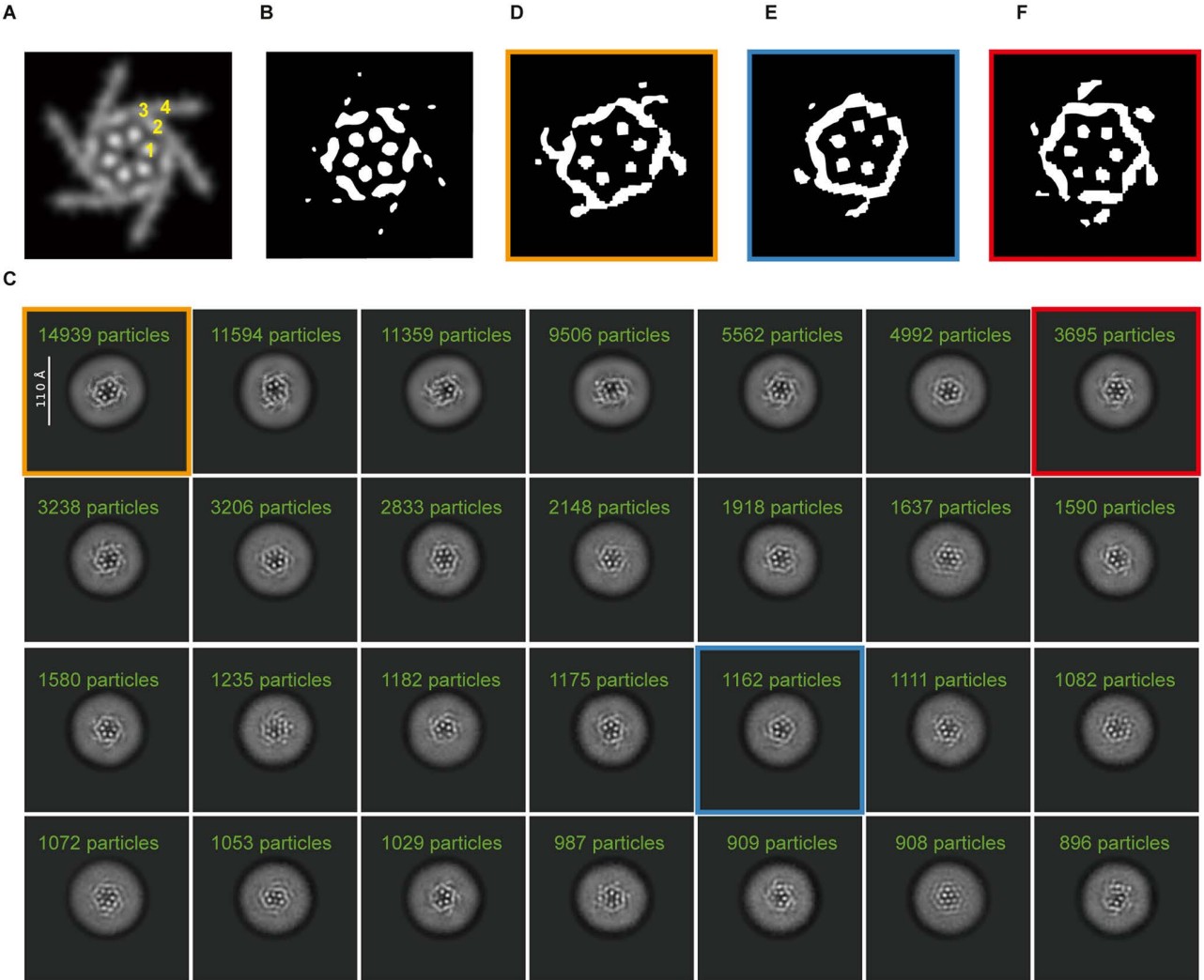

**Fig 2. 2D class averages of hOrai1 in top views.** (A) Simulated EM map of dOrai (generated from PDB ID: 6BBG) low-pass filtered to 8 Å resolution and viewed from the top. The transmembrane helices (M1-M4) are denoted in yellow.(B) The black and white view of (A). (C) Representative 2D class averages of hOrai1 in top views. Selected classes are highlighted with colored boxes: particles with C2 symmetry (orange box), pentameric particles (blue box), and particles with C6 symmetry (red box). (D-F) Binarized black-and-white views of selected 2D class averages boxed in (C).

(Fig 2A-B). The six internal dots correspond to the inner M1 helices, while the edges of the hexagon are formed by tilted M2 and M3 (Fig 2B). The outward-extending spikes represent the M4 helices (Fig 2B). Given the abundance of vertical projections of hOrai1 in the cryo-EM dataset, we selected them for further classification (Fig 2C), which revealed considerable structural heterogeneity. The dominant class averages displayed compressed hexagons with apparent C2 symmetry (Fig 2D). In this class average, we only observed four M4 spikes extending out from the tilted edges, but the other two remaining M4 helices were invisible (Fig 2D), likely due to their flexibility. A smaller subset of particles adopted apparently regular hexagonal shape with all six M4 spikes visible (Fig 2F). In addition, we observed few pentagonal averages (Fig 2E).

During subsequent 3D reconstruction, we identified structural heterogeneity in the sample (Fig 3 and S1 Fig). Using 3D variability analysis [17], we found the hexameric hOrai1 displays continuous motion between conformations showing

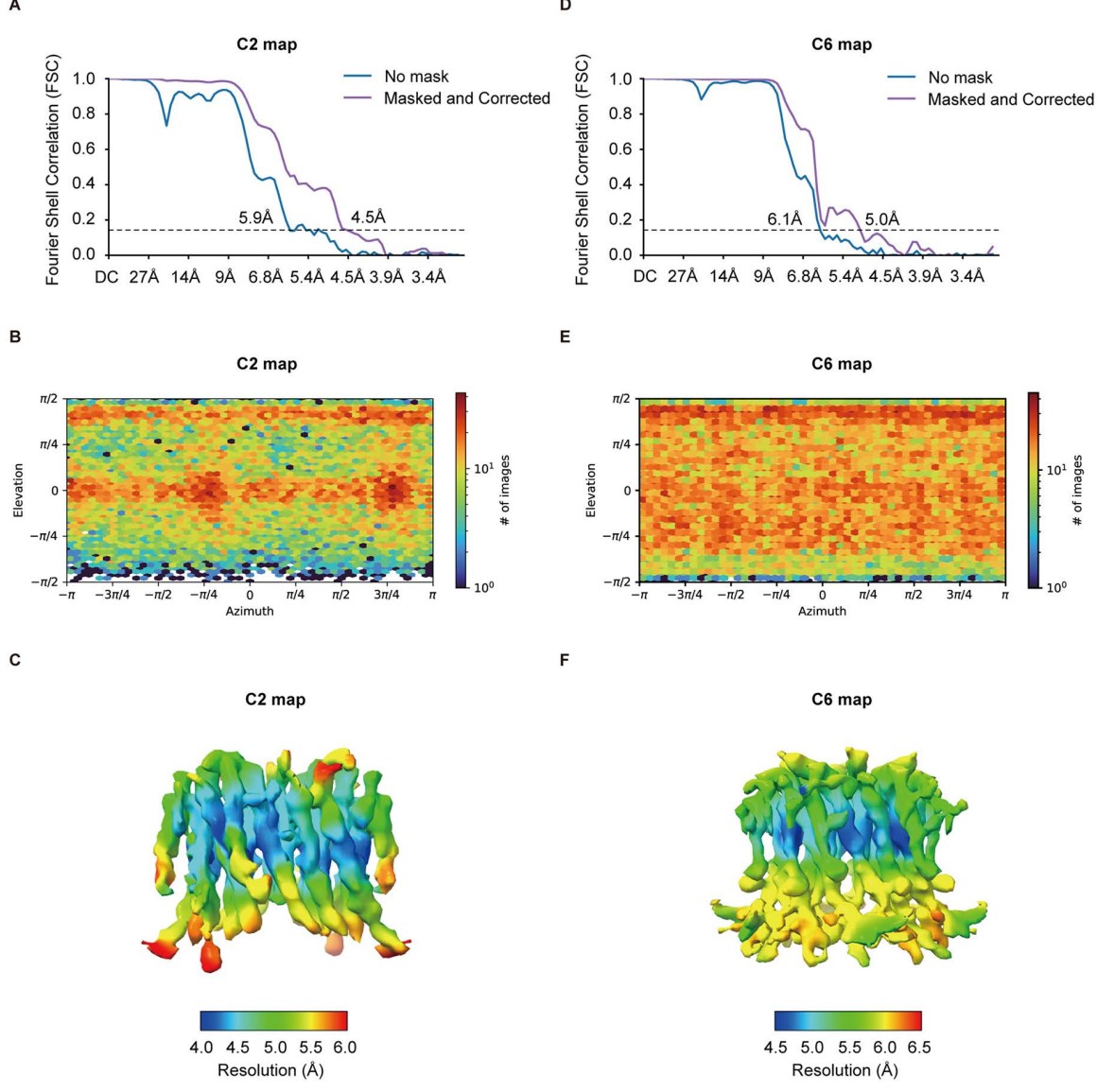

**Fig 3. Cryo-EM reconstruction of hOrai1 with C2 and C6 symmetry.** (A) Gold-standard FSC curve of hOrai1 map with C2 symmetry. Resolution estimations are based on the criterion of FSC = 0.143 cutoff. (B) Particle orientation plot of hOrai1 map with C2 symmetry. (C) Local resolution map of hOrai1 with C2 symmetry. (D) Gold-standard FSC curve of hOrai1 map with C6 symmetry. Resolution estimations are based on the criterion of FSC = 0.143 cutoff. (E) Particle orientation plot of hOrai1 map with C6 symmetry. (F) Local resolution map of hOrai1 with C6 symmetry.

C2 and C6 symmetry (S1 Movie). Further extensive 3D classification and reconstruction yielded two maps: one with C6 symmetry (C6 map) and the other with C2 symmetry (C2 map), at resolutions of 5.0 Å and 4.5 Å, respectively (Fig 3). The presence of a series of intermediate structures between the C2 and C6 maps during 2D and 3D classification suggests that these two maps likely represent two sub-populations within a continuum. Such severe structural heterogeneity prevented us from achieving a higher, near-atomic resolution of hOrai1. Due to the limited map quality, we did not build atomic models and subsequent structural analysis focused on the helical densities in the electron density maps (Fig 4). Comparison of the two maps revealed that in the C2 map, four subunits shifted inward toward the center, while the other

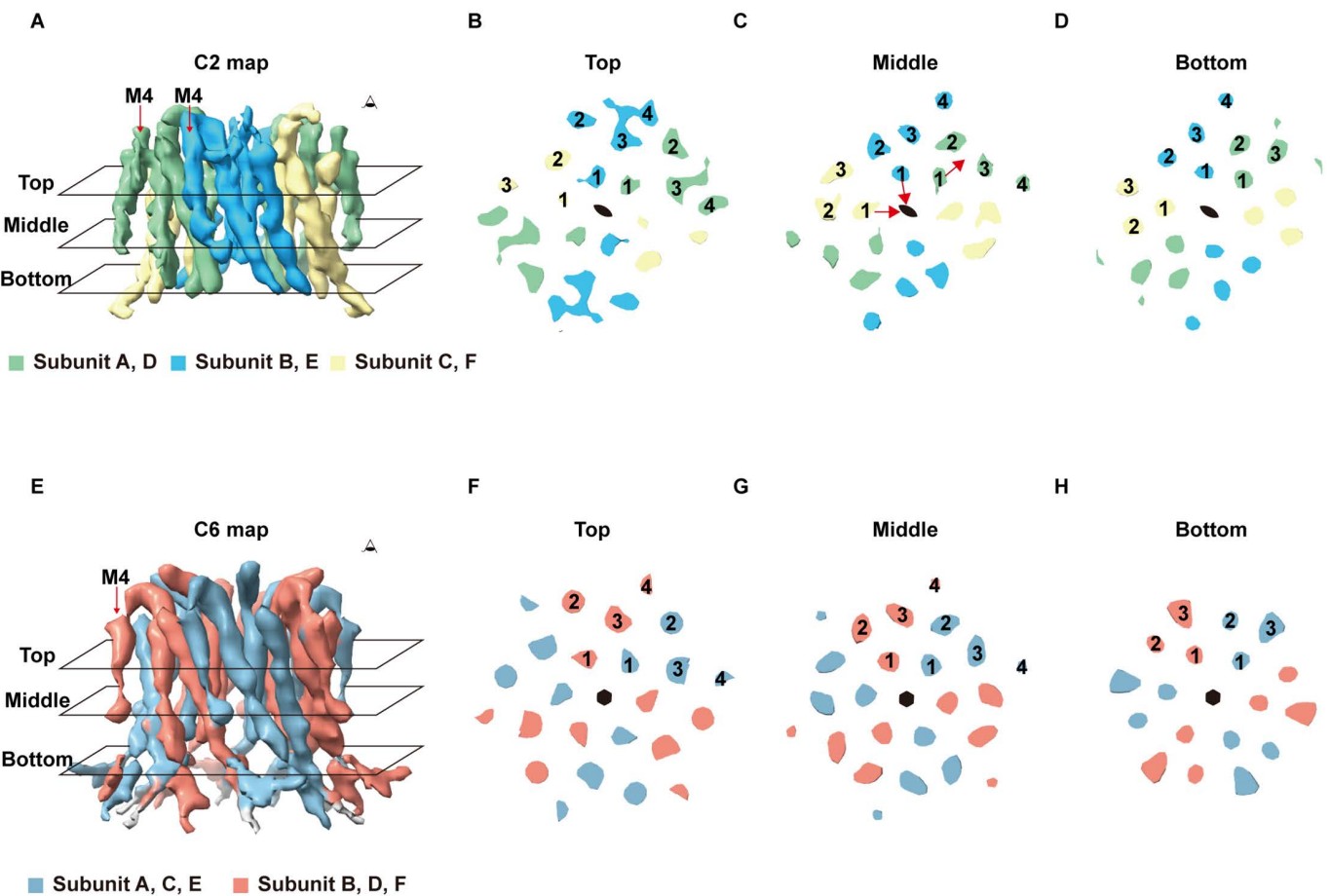

**Fig 4. Medium resolution maps of hOrai1 with C2 and C6 symmetry.** (A) Side view of hOrai1 C2 map. Three subunits within the asymmetric unit are colored in green, blue, and yellow, respectively. The red arrow indicates the M4 helices of hOrai1. (B) Top view of the cross-section of hOrai1 C2 map at the top layer denoted in (A). (C) Top view of the cross-section of hOrai1 C2 map at the middle layer denoted in (A). The red arrow indicates the shift direction of hOrai1 subunits. (D) Top view of the cross-section of hOrai1 C2 map at the bottom layer denoted in (A). (E) Side view of hOrai1 C6 map. Two adjacent subunits are colored in red and blue, respectively. The red arrow indicates the M4 helices of hOrai1. (F) Top view of the cross-section of hOrai1 C6 map at the top layer denoted in (E). (G) Top view of the cross-section of hOrai1 C6 map at the middle layer denoted in (E). (H) Top view of the cross-section of hOrai1 C6 map at the bottom layer denoted in (E).

two moved outward, resulting in an overall compression of the hexamer (Fig 4A-D). The density for the M4 helices was weaker than that of other helices in the sharpened maps, especially for the C6 map (Fig 4E-H). In particular, the density for M4 below the approximate position of P245 was poorly resolved (Fig 4A, 4E, and S1 Fig C-D).

## Map of hOrai1 in the quiescent closed state

We compared the C6 map with previously published structures of dOrai in the unlatched closed state (PDB ID: 6BBG) [7] and the unlatched open state with the H206A mutation (PDB ID: 6BBF) [7]. The C6 map closely resembled the closed conformation of dOrai, with only a slight outward displacement of the M1 helices in hOrai1 (Fig 5A-C). In contrast, the M1 helices underwent a more pronounced outward movement in the open state of dOrai (Fig 5D-F). These observations suggest that the current C6 map of hOrai1 corresponds to the closed conformation.

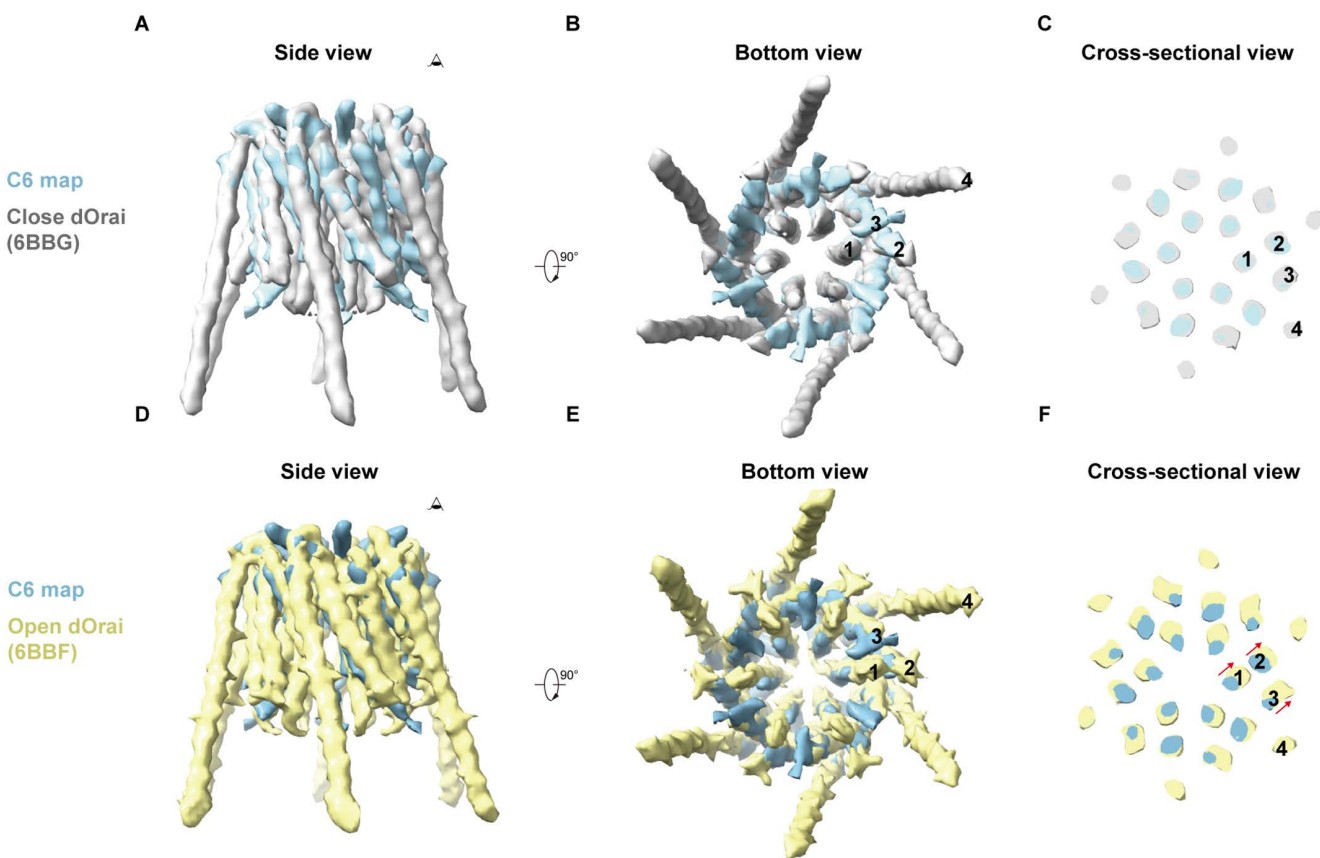

**Fig 5. Comparison of hOrai1 C6 map with dOrai structures.** (A) Side view of comparison of hOrai1 C6 map (cyan) with the simulated map of dOrai at closed state (gray, PDB ID: 6BBG). (B) The bottom view of (A). (C) An enlarged top view of the cross-section in (A). (D) Side view of comparison of hOrai1 C6 map (cyan) with the simulated map of dOrai at open state (yellow, PDB ID: 6BBF). (E) The bottom view of (D). (F) An enlarged top view of the cross-section in (D). The red arrows indicate the relative movement trend of the TM1–3 helices associated with channel opening.

## Discussion

The cryo-EM analysis of hOrai1 purified in detergent micelles revealed that it predominantly assembles as a hexamer. A minor population of pentameric particles was also observed (Fig 2C and E). Pentameric assemblies of hOrai1 have previously been reported as the dominant form in hOrai1–SOAR fusion proteins [10]. This difference in the predominant oligomeric state—hexamer for hOrai1 alone versus pentamer for the fusion construct—may arise because the constitutive presence of SOAR influences complex assembly during maturation. In our dataset, however, the pentamer represents only a minor population, while the hexamer is predominant, suggesting that under the present experimental conditions, the hexamer is the dominant assembly state of hOrai1.

In addition to oligomeric heterogeneity, we observed conformational heterogeneity in the hexameric hOrai1 particles. Specifically, when purified in LMNG/CHS mixed micelles, some hOrai1 adopted a C2 conformation not previously observed in structural studies of dOrai. It remains unclear whether hOrai1 in this C2 conformation exists on the plasma membrane. Notably, we observed a continuous distribution between C2 and C6 conformations, indicating that these states interconvert rather than existing as discrete populations. This continuous structural heterogeneity posed a major technical obstacle to achieving high-resolution reconstruction and hindered accurate model building. At the current resolution (~4.5–5.0 Å), the maps do not permit definitive model building or functional assignment of the observed conformational states.

Importantly, the C2-symmetric particles retain a six-subunit architecture, ruling out partial subunit dissociation or gross misassembly. The presence of intermediate conformations connecting C2 and C6 populations during 2D and 3D classification further supports the idea that these states are interconverting. However, it remains unclear whether the coexistence of C2 and C6 populations reflects a thermodynamic equilibrium of hOrai1, a transient gating-related or inactivated conformation, or a state stabilized by the detergent environment. Although contributions from the detergent cannot be fully excluded, the observation of continuous transitions between C2 and C6 argues against a purely sporadic or damage-related origin. Our results suggest that stabilizing the C6 population through sample optimization or construct modification will be essential for improving map resolution and enabling detailed structural and functional analysis.

## Methods

### Molecular cloning

The hOrai1-ΔN-Strep plasmid was commercially synthesized by BGI Qinglan Biotechnology. The construct features a codon-optimized hOrai1 sequence with an N-terminal 60-amino-acid deletion, a C-terminal Strep-tag II connected by a GGGGGG linker, and was cloned into the pcDNA3.1(+) vector. The final plasmid was verified by full-length sequencing.

### Purification of the hOrai1-ΔN-Strep

Orai1-ΔN-Strep was expressed in suspension HEK293T cells transfected with PEI. Cells were supplemented with 5% FBS at 8 hours post-transfection and harvested at 48 hours. The cell pellet was resuspended in buffer (150 mM NaCl, 25 mM Tris-HCl, 5 mM EDTA, 5 mM EGTA, pH 7.5) and solubilized with 1% LMNG/0.2% CHS at 4 °C for 1 h. The clarified supernatant was subjected to Strep-Tactin affinity purification. After sequential washing with Tris-based (pH 7.5) and MES-based (pH 6.3) buffers containing 0.002% LMNG, the protein was eluted with biotin, concentrated, and applied to a Superose 6 Increase SEC column equilibrated in 25 mM MES, 150 mM NaCl, 0.002% LMNG (pH 6.3). Following concentration of the peak fractions to $A_{280} = 14$ for cryo-EM grid preparation, sample quality was assessed by SDS-PAGE and Native PAGE.

### Cryo-EM sample preparation

Holey carbon grids (Quantifoil Au 300 mesh, R 1.2/1.3) were glow-discharged for 60 s using the Solarus plasma cleaner (Gatan). An aliquot of 2.5 μL of the hOrai1 sample with A280 = 14 was then applied to the grid. The grid was blotted for 2 s using a blot force of −2, and then plunge-frozen into liquid ethane using a Vitrobot Mark IV (Thermo Fisher Scientific).

Cryo-EM grids were screened on the Talos Arctica electron microscope (Thermo Fisher Scientific) operating at 200 kV using a K2 camera (Thermo Fisher Scientific). The images of the screened grid were collected on 300 kV Titan Krios (Thermo Fisher) with a Falcon 4i direct electron camera (Thermo Fisher Scientific). The pixel size was set to 0.75 Å, with a defocus range of −1.0 to −2.0 μm. 16486 image stacks containing 40 frames were collected, with a total dose of 50 e$^-$/Å$^2$.

### Cryo-EM data processing

The image processing workflow is illustrated in S1 Fig. Datasets were first gain-corrected, motion-corrected and dose-weighted by MotionCor2-1.3.2 [18]. CTF estimation for the selected micrographs was performed using GCTF-1.18 [19]. Micrographs with ice or ethane contamination and empty carbon were removed manually. All subsequent picking, classification and reconstruction were performed in cryoSPARC-4.5.3 [20]. A set of 5,000 micrographs was subjected to blob picking, yielding 5,046,391 particles after several rounds of 2D classification. Then the particles of the best class were used for initial model reconstruction and 3D refinement. After seed-facilitated 3D classification [21], the particles of the

best class were selected for Topaz training [22]. 5,369,679 particles were then extracted by Topaz. 3D variability analysis (3DVA) [17] was used to analyze the structural heterogeneity and continuous structural transitions. After several rounds of extensive 2D and 3D classification, the particles with apparent C2 or C6 symmetry were selected and refined separately. The resolution estimation was based on the gold standard FSC 0.143 cut-off.

### Single cell calcium imaging

HEK STIM1-YFP stable cells [11] were cultured at 37 °C with 5% $CO_2$ in DMEM supplemented with 10% FBS, 1% penicillin-streptomycin, and 1 μg/ml puromycin. Transfection was performed by electroporation as previously described [12]. After seeding on coverslips, cells were imaged 24 h later in an imaging buffer containing (in mM): 107 NaCl, 7.2 KCl, 1.2 $MgCl_2$, 11.5 glucose, and 20 HEPES-NaOH (pH 7.2). Fluorescence signals were acquired every 2 s using a ZEISS Observer Z1 system, exported from SlideBook v.6.0.23, and analyzed in MATLAB 2014a to calculate relative changes in R-GECO1.2 fluorescence [12]. Results were plotted using GraphPad Prism 9.51. Data represent mean ± SEM from at least three independent experiments.

## Supporting information

**S1 Fig. Cryo-EM data processing of hOrai1.** (A) Representative 2D class averages. (B) Cryo-EM data processing workflow in cryoSPARC. (C) EM densities for TM4 helix of hOrai1 with C6 symmetry in Fig 4E are contoured at 7.3 σ. The approximate position of the membrane bilayer is indicated by two dashed lines. (D) EM densities for TM4 helices of hOrai1 with C2 symmetry in Fig 4A are contoured at 9.0 σ. Subunit A and B are represented by green and blue separately. (E) EM densities for the TM4 helix of closed dOrai (PDB: 6BBG), with the position of P288 indicated in green. The corresponding amino acid in human Orai1 is shown in parentheses.
(TIF)

**S1 Table. Cryo-EM data collection statistics.**
(DOCX)

**S1 Movie. Continuous structural transition of hOrai1 between C6-like and C2-like conformations.** Movie frames were generated from 3D variability analysis. Only the central section of the transmembrane domain is shown.
(MP4)

**S1 File. Raw images.** Uncropped and unadjusted blot and gel images corresponding to Fig 1C and Fig 1D.
(PDF)

## Acknowledgments

We thank members of the Yihua Huang laboratory for their kind assistance. We are grateful to Keping Hu for providing access to laboratory facilities. Cryo-EM data collection was supported by the Center for Biological Imaging (CBI), Institute of Biophysics, Chinese Academy of Sciences, with the assistance of Xujing Li. Structural computations were performed on the Computing Platform of the Center for Life Sciences and the High Performance Computing Platform of Peking University. We also acknowledge the Experimental Technology Center for Life Sciences, Beijing Normal University, and the National Center for Protein Sciences at Peking University for technical support.

## Author contributions

**Conceptualization:** Youjun Wang, Lei Chen.

**Data curation:** Yiming Zhang, Yuan Wang.

**Formal analysis:** Yiming Zhang, Yuan Wang.

**Investigation:** Yiming Zhang, Jindou Liu, Weiwei Bei, Hongkun Wang.

**Methodology:** Yiming Zhang, Jindou Liu.

**Software:** Yiming Zhang, Yuan Wang, Weiwei Bei.

**Supervision:** Youjun Wang, Junli Wang, Lei Chen.

**Validation:** Yiming Zhang, Yuan Wang.

**Visualization:** Yiming Zhang, Yuan Wang.

**Writing – original draft:** Yiming Zhang, Yuan Wang.

**Writing – review & editing:** Youjun Wang, Lei Chen.

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
