## [Decision Letter · Decision Letter 0]

10 Feb 2026

PONE-D-26-02314Structural dynamics of the human Orai1 channel revealed by cryo-electron microscopyPLOS One

Dear Dr. Wang,

Thank you for submitting your manuscript to PLOS ONE. After careful consideration, we feel that it has merit but does not fully meet PLOS ONE’s publication criteria as it currently stands. Therefore, we invite you to submit a revised version of the manuscript that addresses the points raised during the review process. With respect to these revisions, we ask that you pay particular attention to structure/function questions posed by reviewer 2 and consider the suggested additional analyses to help clarify the findings of this work. In addition, please ensure that the content and quality of figures, and figure captions, is sufficient to clearly communicate your key findings (see reviewer 1 comments).

We look forward to receiving your revised manuscript.

Kind regards,

Jarrod B. French, PhD

Academic Editor

PLOS One

**Journal Requirements:**

Reviewers' comments:

Reviewer's Responses to Questions

**Comments to the Author**

1. Is the manuscript technically sound, and do the data support the conclusions?

Reviewer #1: Yes

Reviewer #2: Yes

2. Has the statistical analysis been performed appropriately and rigorously? 

Reviewer #1: Yes

Reviewer #2: I Don't Know

3. Have the authors made all data underlying the findings in their manuscript fully available?

Reviewer #1: Yes

Reviewer #2: No

4. Is the manuscript presented in an intelligible fashion and written in standard English?

Reviewer #1: Yes

Reviewer #2: Yes

5. Review Comments to the Author

Reviewer #1: This study reports the purification and initial cryo-EM characterization of the human Orai1 (hOrai1) channel, confirming its hexameric assembly and identifying substantial structural heterogeneity—with particles adopting both C6- and C2-symmetric conformations. These findings represent a valuable foundational step for structural investigations of this medically relevant ion channel. That said, the work remains in a preliminary state: the obtained resolution is moderate, the observed structural states are not functionally annotated, and the analysis lacks in-depth interpretation of the conformational dynamics within physiological or disease-relevant contexts. Several critical revisions are required to strengthen the manuscript for potential publication, as outlined below:

1. The core finding of this study is the coexistence of C6- and C2-symmetric hOrai1 particles, indicative of intrinsic structural dynamics. Notably, this very heterogeneity is attributed to the limited resolution (~4.5–5.0 Å), which precludes atomic model building. A major knowledge gap is the complete absence of functional data to assign biological relevance to these distinct conformational states. Key unanswered questions include: Is the C2-symmetric state a gating intermediate en route to channel opening, an inactivated state, or a detergent-induced artifact of sample preparation? The discussion section must be expanded to address these critical questions and contextualize this core finding.

2. Figure quality and accompanying legends require significant improvement. For Fig. 4 (the most analysis-intensive panel, featuring comparisons with dOrai states), the legend is insufficiently descriptive and must explicitly define the meaning of red arrows (e.g., “red arrows indicate the direction of M1 helix movement associated with channel opening”). Additionally, it is recommended that Fig. S1 be moved to the main text as a new Fig. 2 to elevate the presentation of key supporting data.

3. A typographical error is present on Page 4, Line 3: the phrase “We next expressed hOrai1-ΔN-strep and and solubilized it in LMNG-CHS mixed micelles” contains a redundant “and”. This should be corrected to: “We next expressed hOrai1-ΔN-Strep and solubilized it in LMNG-CHS mixed micelles”.

4. A duplication exists in the reference list: Reference #10 and Reference #17 are identical, both citing Lu, X. et al. (2024) Biochem. Biophys. Res. Commun. 733, 150723. This redundancy must be removed, and the reference numbering adjusted accordingly.

Reviewer #2: The authors have resolved for the first time the hOrai1 structure however with weak resolution and not much detailed structural insight. They reveal two resolutions maps C2 and C6, while C6 is the fully symmetric one. Further they perform comparisons with the closed and open dOrai structures and suggest that their hOai1 structure matches more with the closed dOrai structure. While the resolution of hOrai1 structure is of course of big interest in the community, many open questions in terms of structural details remain.

Major:

Is the C2 map a potential physiological state? They authors also mention that they found pentameric populations. Can you further elaborate on the physiological relevance of the different states/maps found?-

Please further comment on the resolution and structure of the pore.

Can the authors comment on the CT in the hOrai1 structures. What is the predicted orientation? Similarly for TM4 – Can the authors predict whether there is kink in TM4?

The authors compare their structure with close dOrai (6BBG) showing fully straightend CT. Why did they chose this structure and not one with the antiparallel crossing CTs? Please also perform a comparison with 4HKR.

6. PLOS authors have the option to publish the peer review history of their article (what does this mean?). If published, this will include your full peer review and any attached files.

Reviewer #1: No

Reviewer #2: No

---

## [Author Response · Author response to Decision Letter 1]

12 Apr 2026

Response to Reviewers’ comments

Please note that our responses to the reviewers’ comments were shown in blue below. Our changes in the manuscript text were shown in red.

Reviewer #1: This study reports the purification and initial cryo-EM characterization of the human Orai1 (hOrai1) channel, confirming its hexameric assembly and identifying substantial structural heterogeneity—with particles adopting both C6- and C2-symmetric conformations. These findings represent a valuable foundational step for structural investigations of this medically relevant ion channel. That said, the work remains in a preliminary state: the obtained resolution is moderate, the observed structural states are not functionally annotated, and the analysis lacks in-depth interpretation of the conformational dynamics within physiological or disease-relevant contexts. Several critical revisions are required to strengthen the manuscript for potential publication, as outlined below:

Specific comments:

1. The core finding of this study is the coexistence of C6- and C2-symmetric hOrai1 particles, indicative of intrinsic structural dynamics. Notably, this very heterogeneity is attributed to the limited resolution (~4.5–5.0 Å), which precludes atomic model building. A major knowledge gap is the complete absence of functional data to assign biological relevance to these distinct conformational states. Key unanswered questions include: Is the C2-symmetric state a gating intermediate en route to channel opening, an inactivated state, or a detergent-induced artifact of sample preparation? The discussion section must be expanded to address these critical questions and contextualize this core finding.

Response:

We thank the reviewer for this insightful comment. To be clear, we think the structural heterogeneity is the reason for the limited resolution. We provided the Supplementary Movie 1 showing the continuous structural heterogeneity in the sample in the revised manuscript. We agree that, at the current resolution (~4.5–5.0 Å), our data do not allow definitive assignment of the observed C6- and C2-symmetric hOrai1 conformations to specific functional states. In the revised Discussion, we have added the following sentences: “Notably, we observed a continuous distribution between C2 and C6 conformations, indicating that these states interconvert rather than existing as discrete populations. This continuous structural heterogeneity posed a major technical obstacle to achieving high-resolution reconstruction and hindered accurate model building. At the current resolution (~4.5–5.0 Å), the maps do not permit definitive model building or functional assignment of the observed conformational states.” we have clarified this limitation and avoided overinterpretation of the biological relevance of these structures.

We also added the following content to the Discussion section: “Importantly, the C2-symmetric particles retain a six-subunit architecture, ruling out partial subunit dissociation or gross misassembly. The presence of intermediate conformations connecting C2 and C6 populations during 2D and 3D classification further supports the idea that these states are interconverting. However, it remains unclear whether the coexistence of C2 and C6 populations reflects a thermodynamic equilibrium of hOrai1, a transient gating-related or inactivated conformation, or a state stabilized by the detergent environment. Although contributions from the detergent cannot be fully excluded, the observation of continuous transitions between C2 and C6 argues against a purely sporadic or damage-related origin. Our results suggest that stabilizing the C6 population through sample optimization or construct modification will be essential for improving map resolution and enabling detailed structural and functional analysis.”. We further explicitly state that the C2-symmetric particles cannot be conclusively classified as “a transient gating-related or inactivated conformation, or a state stabilized by the detergent environment”.

Consistent with the scope of PLOS ONE, we present these observations as structural evidence of conformational variability without assigning a specific physiological role. We have revised the Discussion accordingly to clearly delineate the limits of interpretation and to outline future approaches required to establish functional relevance.

2-1. Figure quality and accompanying legends require significant improvement.

Response:

We thank the reviewer for this helpful suggestion. To enhance clarity and professionalism, we have reprocessed all images at high resolution, standardized the font sizes, and improved the color contrast for better visibility. Furthermore, we have substantially revised all figure legends to ensure they are clear.

2-2. For Fig. 4 (the most analysis-intensive panel, featuring comparisons with dOrai states), the legend is insufficiently descriptive and must explicitly define the meaning of red arrows (e.g., “red arrows indicate the direction of M1 helix movement associated with channel opening”).

Response:

We thank the reviewer for the reminder. We have specified in the figure legend that “The red arrows indicate the relative movement trend of the TM1-3 helices associated with channel opening.”

2-3. Additionally, it is recommended that Fig. S1 be moved to the main text as a new Fig. 2 to elevate the presentation of key supporting data.

Response:

We thank the reviewer for the suggestion. We have moved some panels in Fig. S1 to Fig. 3 and revised the numbering of all figures in the manuscript.

3. A typographical error is present on Page 4, Line 3: the phrase “We next expressed hOrai1-ΔN-strep and and solubilized it in LMNG-CHS mixed micelles” contains a redundant “and”. This should be corrected to: “We next expressed hOrai1-ΔN-Strep and solubilized it in LMNG-CHS mixed micelles”.

Response:

We thank the reviewer for the correction. The text has been updated as follows: “We next expressed hOrai1-ΔN-strep and solubilized it in LMNG-CHS mixed micelles. The protein was purified via Streptactin affinity chromatography and size-exclusion chromatography (SEC), which showed a monodisperse peak (Fig. 1B). ”

4. A duplication exists in the reference list: Reference #10 and Reference #17 are identical, both citing Lu, X. et al. (2024) Biochem. Biophys. Res. Commun. 733, 150723. This redundancy must be removed, and the reference numbering adjusted accordingly.

Response:

We thank the reviewer for noting this, and the error has been corrected.

Reviewer #2:

The authors have resolved for the first time the hOrai1 structure however with weak resolution and not much detailed structural insight. They reveal two resolutions maps C2 and C6, while C6 is the fully symmetric one. Further they perform comparisons with the closed and open dOrai structures and suggest that their hOai1 structure matches more with the closed dOrai structure. While the resolution of hOrai1 structure is of course of big interest in the community, many open questions in terms of structural details remain.

Major:

1. Is the C2 map a potential physiological state? They authors also mention that they found pentameric populations. Can you further elaborate on the physiological relevance of the different states/maps found?

Response:

We thank the reviewer for this important question. At the current resolution, we cannot definitively assign the C2-symmetric map to a specific physiological state. In the revised Discussion, we clarify that the C2 population may represent one of several possibilities, including a thermodynamically accessible conformational state, a transient gating- or inactivation-related conformation, or a state stabilized under detergent conditions. We emphasize that no definitive functional assignment is made. And added the following statement in the Discussion: "At the current resolution (~4.5–5.0 Å), the maps do not permit definitive model building or functional assignment of the observed conformational states."

Importantly, the C2-symmetric particles retain a complete hexameric architecture and appear in a continuum with the C6 population during 2D and 3D classification, suggesting reproducible conformational variability rather than sporadic misassembly. We have added the following discussion in the corresponding section: "Importantly, the C2-symmetric particles retain a six-subunit architecture, ruling out partial subunit dissociation or gross misassembly. The presence of intermediate conformations connecting C2 and C6 populations during 2D and 3D classification further supports the idea that these states are interconverting."

Regarding the pentameric population of hOrai1, we note that pentameric assemblies of hOrai1 have previously been reported in hOrai1–SOAR fusion constructs. We have added the following statement in the Discussion section: "In our dataset, however, the pentamer represents only a minor population, while the hexamer is predominant, suggesting that under the present experimental conditions, the hexamer is the dominant assembly state of hOrai1." We speculated the pentameric species are assembly intermediate of hOrai hexamer.

2. Please further comment on the resolution and structure of the pore.

Response:

The moderate resolution in the current maps limits detailed structural interpretation of the pore. In the revised manuscript, we clarify that the density corresponding to the pore-lining M1 helices is clearly identifiable and allows assignment of the overall pore architecture, but the map quality does not support reliable side-chain positioning or precise measurement of pore radius. We therefore refrain from detailed mechanistic interpretation of ion permeation. The text has been revised to “pore-lining helices” to better reflect these limitations.

3. Can the authors comment on the CT in the hOrai1 structures. What is the predicted orientation? Similarly for TM4 – Can the authors predict whether there is kink in TM4?

Response:

We have supplemented additional Supplementary Fig. 1C-D to show the densities of the M4 helices and CT. As a reference, we also provided the dOrai structure in the revised Supplementary Fig. 1E. At the current resolution, the density corresponding to the C-terminal region is relatively weak and partially discontinuous, precluding definitive modeling of its structures.

4. The authors compare their structure with close dOrai (6BBG) showing fully straightend CT. Why did they chose this structure and not one with the antiparallel crossing CTs? Please also perform a comparison with 4HKR.

Response:

The M4 helix in 4HKR adopts a latched conformation not observed in any other dOrai structures, including 6BGG, 6BBF, 6BBH, 6BBI, 6AKI, 7KR5. Examination of the crystal packing in 4HKR reveals that the latched M4 is heavily involved in crystal contacts (Fig. 1). Thus, we speculate that the latched M4 conformation in 4HKR may result from crystal packing and may not be suitable for structural comparison with hOrai1.

Fig. 1 Crystal structure of dOrai (PDB: 4HKR). Three dOrai hexamers are shown in gray, with their TM4 helices involved in crystal packing highlighted in yellow, purple, and green, respectively.

---

## [Editor Report · Decision Letter 1]

16 Apr 2026

Structural dynamics of the human Orai1 channel revealed by cryo-electron microscopy

PONE-D-26-02314R1

Dear Dr. Wang,

We’re pleased to inform you that your manuscript has been judged scientifically suitable for publication and will be formally accepted for publication once it meets all outstanding technical requirements.

Kind regards,

Jarrod B. French, PhD

Academic Editor

PLOS One

---

## [Editor Report · Acceptance letter]

PONE-D-26-02314R1

PLOS One

Dear Dr. Wang,

I'm pleased to inform you that your manuscript has been deemed suitable for publication in PLOS One. Congratulations! Your manuscript is now being handed over to our production team.

Kind regards,

on behalf of

Professor Jarrod B. French

Academic Editor

PLOS One